# Hydrodynamic instability at impact interfaces and planetary implications

Avi Ravid [1,2], Robert I. Citron [1] & Raymond Jeanloz [1✉]

Impact-induced mixing between bolide and target is fundamental to the geochemical evolution of a growing planet, yet aside from local mixing due to jetting – associated with large angles of incidence between impacting surfaces – mixing during planetary impacts is poorly understood. Here we describe a dynamic instability of the surface between impacting materials, showing that a region of mixing grows between two media having even minimal initial topography. This additional cause of impact-induced mixing is related to Richtmyer-Meshkov instability (RMI), and results from pressure perturbations amplified by shock-wave refraction through the corrugated interface between impactor and target. However, unlike RMI, this new impact-induced instability appears even if the bodies are made of the same material. Hydrocode simulations illustrate the growth of this mixing zone for planetary impacts, and predict results suitable for experimental validation in the laboratory. This form of impact mixing may be relevant to the formation of stony-iron and other meteorites.

[1] UC, Berkeley, CA, USA. [2] Soreq NRC, Yavne, Israel. ✉email: jeanloz@berkeley.edu

Mixing during planetary impact remains poorly understood, as illustrated by ongoing controversy regarding the origin of the Moon and its composition relative to that of Earth[1,2], and the formation and abundance of stony iron meteorites[3]. Jetting, essentially a kinematic effect between surfaces colliding at high angles, is well known[4,5]. Mixing processes associated with more general impact conditions warrant further characterization, however.

Here we consider a dynamic instability of the interface between impacting media having distinct impedances $Z = \rho_0 U_S$ ($\rho_0$ is initial density and $U_S$ is shock velocity), when one or both have topography (Fig. 1). The initial topography can be gentle, and we take it to be

$$X(y) = a_0 e^{iky}, a_0 \ll \lambda = 2\pi/k \qquad (1)$$

with $y$ being perpendicular to the impact direction $x$; the general case is three-dimensional, but for simplicity our discussion is in two-dimensions. The vorticity generated due to the corrugated initial interface causes the initial topography to grow over time, resulting in enhanced mixing. For jetting, in contrast, the angle of incidence must exceed a critical value, determined by the material properties and impact velocity[4]. All of the examples presented here are within the jet-free regime, at small angles of incidence.

While the amplitude continues to grow, another mixing mechanism appears as the initially smooth interface between impactor and target becomes turbulent due to shear (Kelvin–Helmholtz instability), evident as small ripples on the iron-dunite interface in Fig. 1[6].

The loading paths and geometry are summarized in Fig. 2. In order to facilitate comparison with the Richtmyer–Meshkov instability (RMI), we choose the impact velocity $U_{impact}$ such that the post-impact shock pressure in the high-impedance material is the same for the different simulations (Fig. 2a, point 2 on the Hugoniot). This places the low-impedance material in a slightly different thermodynamic state for the two cases, however, due to the different loading paths (see Supplementary Information). We ignore strength effects, thereby assuming that the shocked materials are well above the Hugoniot elastic limit and behave as fluids. Figure 2b illustrates the tilting of the shock fronts relative to both the individual (initial) and joint material interfaces, for a distinct point on the wavy surface, which defines the local angle of impact ($\alpha_1$).

For the wavy surface, the local impact angles vary along the $y$-axis maintaining $|\alpha_1| \leq 2\pi \cdot a_0/\lambda$. The velocities of the shocks propagating through unshocked low- and high-impedance materials are $U_1$ and $U_2$, respectively; $U_3$ is the velocity of the interface between them. From mass and momentum conservation

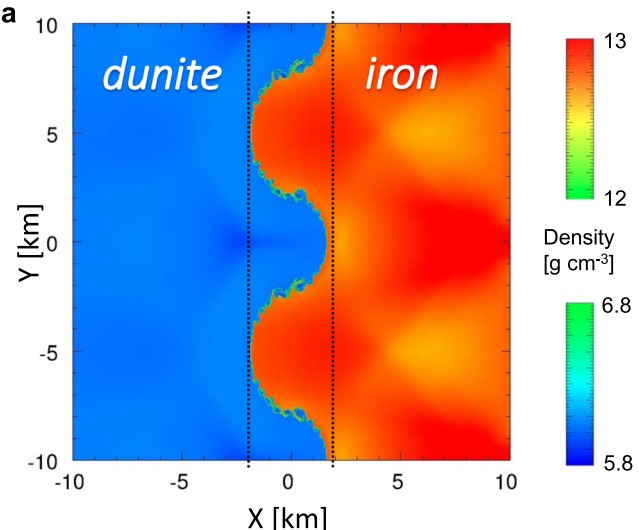

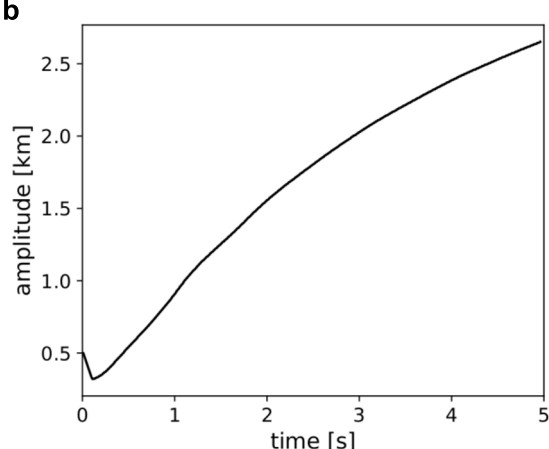

**Fig. 1 Large scale dunite-iron impact induces interface instability. a** Map of densities of a corrugated iron plate (red) impacting a flat dunite target (blue) at $U_{impact} = 10.5$ km/s, shown 2.39 s after impact, as calculated using the CTH code[9]. The potential mixing zone is marked between dashed lines. The initial topography has amplitude $a_0 = 0.5$ km and wavelength $\lambda = 10$ km, and the model domain extends $2\lambda = 20$ km in the perpendicular ($y$) direction. **b** Calculated amplitude of the interface between iron and dunite over time. After a brief initial compression with a minimal amplitude of $a_0(1 - U_{particle}/U_{impact})$, the interface amplitude increases monotonically, thereby enlarging the potential mixing zone ($U_{particle}$ is particle velocity). As the interface does not remain symmetrical with time, its amplitude is taken to be $a(t) = (max_{crest} - min_{trough})/2$.

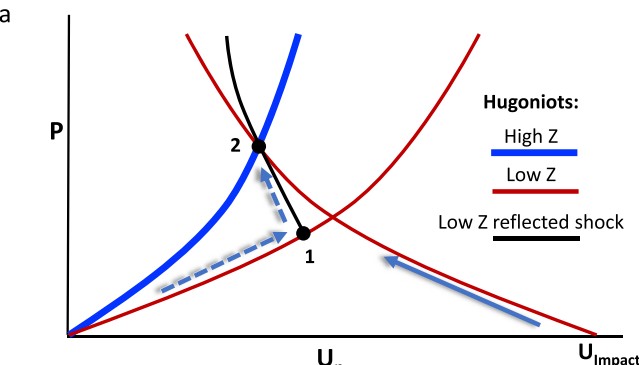

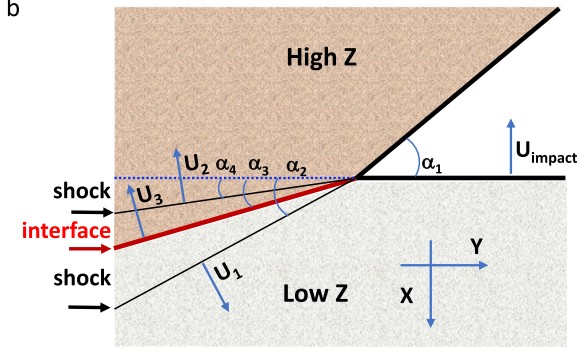

**Fig. 2 Material loading paths, impact vs. shock. a** Two paths are considered to reach the shocked state (2) of the high-impedance material (blue curve): (i) impact by a lower-impedance material at impact velocity $U_{impact}$ (solid arrow); or (ii) reflected shock through point 1 (dashed arrows) **b** Tilting of the shock fronts and interface are illustrated for a projectile with velocity ($U_{impact}$) impacting a wavy surface target at a local oblique angle ($\alpha_1$). All angles are defined to be positive.

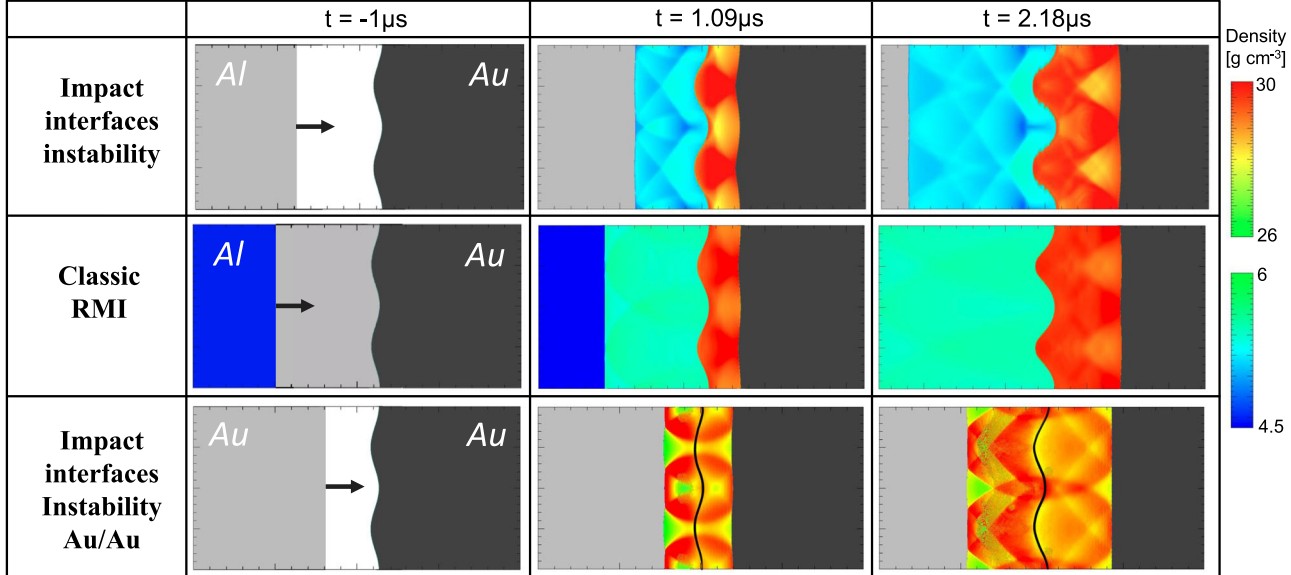

**Fig. 3 Three cases of interface instability.** Comparison between the interface instability following an impact in which the projectile or target has topography (top) and the classical Richtymer–Meshkov instability (RMI) for a shock transmitted through a corrugated interface (middle, shock moving from left to right), as simulated for aluminum and gold using the CTH code[9]. Two density ranges are plotted to highlight the wave interferences behind the shock fronts in the two materials (scales on right), with the dimensions being 4 cm in the direction of impact ($x$) and 2 cm in the orthogonal direction ($y$) (the calculation itself extends from −15 to +15 cm in the $x$-direction). In the present case of impact-induced instability, forward- and backward-propagating shocks are generated at $t = 0$ by collision of a flat Al plate (gray: initial density) and a corrugated Au surface (black: initial density). Either medium can be impactor or target, due to symmetry discussed in the text. For the classical RMI, a shock generated up-range arrives at $t = 0$ at the wavy Al/Au interface, producing forward- and backward-propagating shocks. In both cases, the interface amplitude increases with time, indicative of a growing zone of mixing between impactor and projectile (Fig. 4). Light gray, black, and blue indicate densities of unshocked Al, unshocked Au, and (for classical RMI) initially shocked Al, respectively. The special case of impact between identical materials (Au/Au) is presented at the bottom (interface marked by a black line); there is no classical RMI analogy for this case involving identical projectile and target materials.

(Hugoniot relations), the shock pressure $P$ and volume change $\Delta V$ are accompanied by a jump in material velocity normal to the shock front. Assuming zero initial pressure, and starting with zero velocity in the high-$Z$ material along the $x$-axis, we obtain the following material velocities after impact (point no. 2 in Fig. 2a):

$$u_Y^H = -\sqrt{P_2(V_0^H - V_2^H)} \cdot sin(\alpha_4)$$
$$u_Y^L = \sqrt{P_2(V_0^L - V_2^L)} \cdot sin(\alpha_2)$$
$$u_X^H = -\sqrt{P_2(V_0^H - V_2^H)} \cdot cos(\alpha_4)$$
$$u_X^L = U_{impact} + \sqrt{P_2(V_0^L - V_2^L)} \cdot cos(\alpha_2)$$
(2)

Superscripts $L$ and $H$ refer to low- and high-impedance materials, respectively, and $V$ is specific volume. The generation of transverse material velocities, $u_y$, is the root cause of the emerging instability of the interface, through generation of pressure perturbations and vorticity.

Our analysis is similar to that of Richtmyer and Meshkov for the instability of a shock propagating through the wavy interface between two fluids[7,8]. Indeed, (2) corresponds to Richtmyer's Equation (44)[7].

Unlike the classical RMI, however, the present case is symmetrical because the roles of the projectile and target can be reversed (i.e., the velocities can be taken in either—or even a different—frame of reference). The initial topography can be on the impactor, the target or both. Also, due to its geometric origin, the impact-induced instability appears even if the impacting bodies are made of the same material, as discussed below.

**Results and discussion**
We take a closer look at the impact-induced instability to compare it with the RMI by considering aluminum and gold, two

materials with well-known shock properties (Fig. 3). The shock pressure (Fig. 2a, point 2 on the Hugoniot) is ~300 GPa, well above the shock melting pressure of both materials. The sample size and settings are within the reach of contemporary experimental systems, such as two-stage light gas-guns.

We simulate the impact-induced interface instability using the numerical hydrocode CTH[9] (see Methods), and model impacts of a flat aluminum plate onto a corrugated gold plate (high-$Z$/low-$Z$ collision), and of a flat gold plate onto a corrugated gold plate (identical-$Z$ collision). The initial amplitude of the wavy surface is 0.5 mm, and the initial wavelength is 1 cm (the simulation domain extends two wavelengths in the $y$ direction). The impact velocities are 9.5 km/s for the Al/Au case, and 4.6 km/s for Au/Au. In order to hold the center of the Al/Au mixing zone fixed in our laboratory frame, the gold velocity is 2.3 km/s to the left, and the aluminum 7.2 km/s to the right.

For comparison we also model a classical RMI, in which the gold and aluminum plates are initially in contact along a wavy interface. A shock wave is generated up-range of the interface by impacting the flat end of the aluminum plate with another flat aluminum plate, so that the shock wave yields the same peak pressure on the gold target when it passes through the interface.

In Fig. 3 we use two density scales in order to highlight the perturbations in density (pressure and temperature, as well). The amplitude growth rate (Fig. 4a) displays a similar structure in both cases, with a slightly faster growth for the impact-interface relative to the RMI.

The gold-gold impact case is interesting, in that the amplitude growth is evident, but without the double-peak structure in Fig. 4b that comes from the pressure perturbations passing through and reflecting off each other due to impedance mismatch

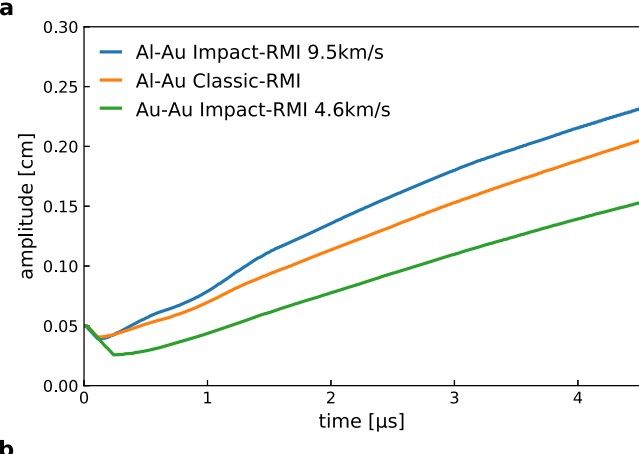

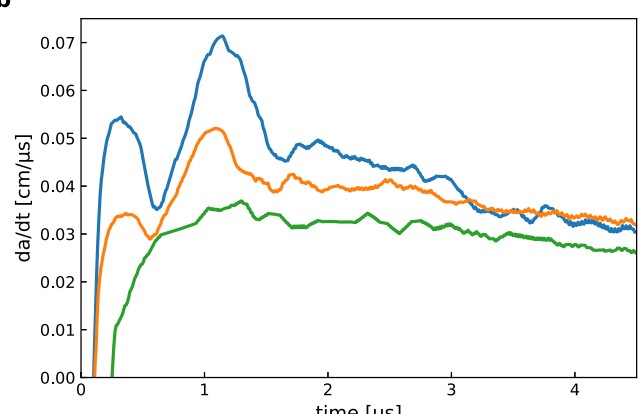

**Fig. 4 Interface instability growth. a** Amplitude vs. time, $a(t)$, for three cases with final pressure of ~300 GPa: Al-Au, interface instability (blue); Al-Au, classical RMI (red); and Au-Au interface instability (green). **b** Growth rate ($da/dt$ vs. time) for the same three cases.

at the contact layer. There is no interface, hence no instability for the Au-Au Richtmyer–Meshkov case.

The implication of this last point is that planetary impact between materials having similar or even identical impedance (e.g., similar compositions and porosity) can result in the vorticity and consequent mixing associated with the interface instability discussed here. This is important in the case of impactor and target having distinct isotopic or trace-element signatures, as mixing becomes possible between mechanically similar but geochemically distinguishable materials. Energy dissipation is also expected in connection with this vorticity.

Along with the impact-induced interface instability, additional vorticity can emerge due to Rayleigh–Taylor and Kelvin–Helmholtz instabilities, depending on the impact conditions and geometry (e.g., Fig. 1)[10,11]. These different instabilities are controlled by distinct dynamics, so in general occur at different length scales, all the more so for a wide spectrum of topographic wavenumbers. The resulting multi-scale turbulent mixing can increase both the pervasiveness and heterogeneity of the blending of the two media[12–14].

We have demonstrated that for high-velocity impacts, a dynamical instability of the interface between the two media generates a growing zone of potential mixing between impactor and target, even for gentle initial topography with amplitude much smaller than the wavelength. This instability arises whether or not the densities and impedances of the two colliding bodies are identical. For this reason, as well as the symmetry between impactor and target, the process we describe is distinct from RMI[15].

Scaling our simulations using an impulsive model shows that impacts at planetary-relevant velocities on the order of 10 to 30 km/s result in mixing zones 10–20% the size of the shocked region (see Supplementary Information). This implies that a significant amount of mixed material is generated early in the impact process, even for gentle initial topography. Moreover, porosity tends –if anything—to enhance the relative size of the interface instability mixing zone, which is of interest because small planetary bodies tend to have considerable porosity.

Finally, impact-induced interface instability may help explain mixed compositions and textures observed in certain meteorites[5,16]. For example, collisional mixing of protoplanetary mantle and core materials is thought to explain the formation and abundance of stony-iron meteorites[17–19]. Overall, however, the process we describe ought to be considered over a broad range of scales for planetary-impact models that aspire to explain geochemical similarities or differences between remnant objects, as collisional mixing between planetesimals and asteroids of similar rheology but distinct composition appears to be an important process in explaining the range of compositions for planetesimals and meteorite parent bodies[19–22].

## Methods

We use CTH version 11.1 (McGlaun et al. Schmitt et al.)[9,23]. All simulations are conducted in 2D using adaptive mesh refinement to refine the mesh at material and shock interfaces (Crawford)[24]. The maximum resolution of the adaptively refined mesh is 1280 cells per wavelength. We use reflective (periodic) boundaries at the top and bottom of the domain. Both the impacting plate and target plate are given a negative and positive velocity, respectively, so that the interface stays near $x = 0$ for the simulation duration.

Gold and aluminum are modeled using the Mie–Grüneisen equation of state with the default parameters from CTH. For gold, we use an initial density $\rho_0 = 19.24$ g/cm$^3$, bulk sound speed $c_s = 3.056$ km/s, Hugoniot slope $S_1 = 1.572$, Grüneisen parameter $\gamma_0 = 2.97$ and specific heat $C_V = 1.47$ kJ/g/eV. For aluminum, we use the parameters for Al-1100: $\rho_0 = 2.707$ g/cm$^3$, $c_s = 5.25$ km/s, $S_1 = 1.37$, $\gamma_0 = 1.97$, and $C_V = 10.7$ kJ/g/eV. Iron and dunite are modeled using the ANEOS equation of state with iron parameters from CTH and dunite parameters from Canup et al. (2013)[25].

The amplitude of the interface was computed by taking the distance between furthest cells with a partial volume fraction of material. The x-location of the interface within a mixed material cell was computed based on the fraction of the cell width equal to the volume fraction of material in the cell.

Hyperlinked in the supplementary information are two short videos of the Gold-Gold (Supplementary Movie 1) and Aluminum-Gold (Supplementary Movie 2), impact cases, as described in the main text and Fig. 4. The videos display the densities and materials during the process. In order to make the amplitude growth easier to track, the original topography of the materials is highlighted with a solid line.

## Data availability

The authors confirm that the data supporting the findings of this study are available within the article and its supplementary materials.

## Code availability

The CTH code is licensed by Sandia National Laboratories and is not publicly available. The data output used to generate the figures included in this manuscript is available on request.

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

## Acknowledgements

We thank G. W. Collins (U Rochester), M. Manga (UC Berkeley), D. D. Meyerhofer (Los Alamos), S. Stewart (UC Davis), S. Eliezer (Soreq NRC), and P. D. Asimow (Caltech) for helpful discussions, and acknowledge support from NSF CMAP (PHY-2020249) and EAR 16152, and from CMEC (NNSA DE-NA0003842).

## Author contributions

A.R. and R.J. conceived the project and developed the theory and R.I.C. performed the numerical simulations. All three wrote the paper.

## Competing interests

The authors declare no competing interests.
