## [Peer Review File · Nature Communications]

Reviewers' Comments:

Reviewer #1:

Remarks to the Author:

Review of Ravid, Citron & Jeanloz NCOMMS Ms

This is a nice, simple, and novel result. It will be of interest to the planetary science community. In fact, it is of interest to me because I have experimental data already collected that has been sitting on my shelf for 10 years waiting for a theoretical framework with which to interpret it. I think it is suitable for publication in Nature Communications after minor revision. My main comment concerns Figure 3.

Gratuitous extra "is" in the sentence "The vorticity is generated due to the corrugated initial interface causes the initial topography to grow over time..."

I don't understand the sentence "In order to make the comparison easier, we choose the impact velocity (U_{impact} , traveling in the $-x$ direction) such that the post-impact shock pressure in the high-impedance (high- Z) material equals the pressure obtained by the shock passing through the interface (Fig. 2a, point 2 on the Hugoniot)". Isn't this always true, for any value of U_{impact} ?

Missing "to" in "Indeed, (2) corresponds Richtmyer's Equation (44)".

Figure 3 introduces (for me) considerable confusion because it shows only two cases whereas the text discusses three cases. Worse, the text starts in one paragraph by discussing two cases (one shown in Figure 3, one NOT shown in Figure 3) and then introduces the third case (shown in Figure 3) in a subsequent paragraph. I strongly advise adding an illustration of the Au-Au impact instability calculation to Figure 3, so that the reader doesn't lose the flow of the argument at this point!

It seems to me that it is worth explaining the origin of the initial transient when the amplitude shrinks. I take it this is just volume compression of the materials and hence passive compression of the interface topography upon initial shock ... but if so (or if not) the reader may want to know that.

In the Al-Au video, it might be good to add a marker of the initial boundary position, to better visualize its growth relative to the initial amplitude.

To a non-geochemist, the significance of mixing between identical materials might be obscure. Who cares if Au mixes with Au, if they are both Au? Well, of course in nature we have trace elements and isotopic signatures that do not affect the impedance but still provide critical tracers of material mixing. So, when two basically similar chondritic or iron asteroids collide, it is important to know the extent of material mixing that occurs, even though both materials have similar or effectively identical shock impedance. I think this point is worth making.

Prepared by Paul Asimow.

Reviewer #2:

Remarks to the Author:

I don't see any specific boxes for the recommendation on this page, so I will include it here:

* Recommend publication with minor revision

General comments:

I do think this is a worthwhile paper that is of interest and has novel results and is worth publishing. It does draw attention to a physical effect that could have implications for planetary science and the formation and composition of asteroids.

The analysis of the phenomenon (a quasi-Richtmyer-Meshkov-like hydrodynamic instability is not very detailed, but further work could come in a later paper. I am somewhat puzzled about the persistence of the instability as found by modeling the collision of impactor and target when they are made of the same material (as in the Au-Au collision simulation). It seems to me that this is material dependent: would it occur in a perfect gas? (As is seen in the original Richtmyer-Meshkov instability). I should think not, not least as a perfect gas would not maintain a surface with a vacuum interface, unlike a solid or liquid.

On page 6 I am not sure I understand the reasoning behind the choice of setting the post-impact Atwood number At_+ to 1. Why is that value chosen? If the materials are the same, then why is ρ_{02+} different from ρ_{01+} ? Or are they somehow different due to the obliqueness of the shocks?

Comments about the text:

P1: "The vorticity is generated due to the corrugated initial interface causes the initial topography to grow over time" -- the sentence is unclear, I suggest inserting the word "that" before "causes..."

P2: "This place the low-impedance material" -- should be "places" I think.

P6: The wording "sizes of shocked zone" is confusing since it seems to refer to a velocity difference (ΔU_s); "sizes" and "perturbation zone" makes me think the authors are talking about physical sizes (distances). Reading it over, it becomes even less clear to me: "relative mixing-zone size":

What is the physical region that the authors are referring to?
Relative to what? Impactor diameter?

I suggest wording something like

"By comparing the magnitudes of the velocity difference ΔU_s (derived from the Hugoniot relations) and the perturbation velocity $V_{prt} = 2 da/dt$, we derive an estimate of the mixing efficiency "

or

"...we derive an estimate of the size of the region over which mixing occurs relative to the impactor size"

if that is what is meant.

The addressed points are highlighted in blue color in the revised manuscript.

Authors' response to Paul Asimow comments:

The authors are grateful for the inspiring and helpful comments.

1. *Gratuitous extra "is" in the sentence "The vorticity is generated due to the corrugated initial interface causes the initial topography to grow over time..."*

Authors – we fixed this typo in the manuscript (p. 1).

2. *I don't understand the sentence "In order to make the comparison easier, we choose the impact velocity (U_{impact} , traveling in the $-x$ direction) such that the post-impact shock pressure in the high-impedance (high-Z) material equals the pressure obtained by the shock passing through the interface (Fig. 2a, point 2 on the Hugoniot)". Isn't this always true, for any value of U_{impact} ?*

Authors – Our intention was to explain that we keep the post shock or post impact state of the High-Z material identical both for the "classical" Richtmyer-Meshkov instability case and our Hydrodynamic instability at impact interfaces. Thus, making the comparison of the different cases easier to understand. Paul Asimow comment indicates that we need a better explanation. We rephrased the sentence in the manuscript, in order to make the statement clearer to the readers (p. 2).

3. *Missing "to" in "Indeed, (2) corresponds Richtmyer's Equation (44)".*

Authors – we fixed this typo in the manuscript (p. 2).

4. *Figure 3 introduces (for me) considerable confusion because it shows only two cases whereas the text discusses three cases. Worse, the text starts in one paragraph by discussing two cases (one shown in Figure 3, one NOT shown in Figure 3) and then introduces the third case (shown in Figure 3) in a subsequent paragraph. I strongly advise adding an illustration of the Au-Au impact instability calculation to Figure 3, so that the reader doesn't lose the flow of the argument at this point!*

Authors – As suggested, we added the Au-Au impact instability case to Fig. 3 (p. 3).

5. *It seems to me that it is worth explaining the origin of the initial transient when the amplitude shrinks. I take it this is just volume compression of the materials and hence passive compression of the interface topography upon initial shock ... but if so (or if not) the reader may want to know that.*

Authors – As suggested, an explanation is placed under Fig. 1 (p. 1)

6. *In the Al-Au video, it might be good to add a marker of the initial boundary position, to better visualize its growth relative to the initial amplitude.*

Authors – We agree and have done as the referee suggests.

7. *To a non-geochemist, the significance of mixing between identical materials might be obscure. Who cares if Au mixes with Au, if they are both Au? Well, of course in nature we have trace elements and isotopic signatures that do not affect the impedance but still provide critical tracers of material mixing. So, when two basically similar chondritic or iron asteroids collide, it is important to know the extent of material mixing that occurs, even though both materials have similar or effectively identical shock impedance. I think this point is worth making.*

Authors – This point is exactly correct, and we have made it more explicitly in the 1st paragraph on p. 4 of the revised manuscript.

Authors' response to Reviewer #2 comments:

Again, the authors are grateful for the inspiring and hopeful comments.

1. *I am somewhat puzzled about the persistence of the instability as found by modeling the collision of impactor and target when they are made of the same material (as in the Au-Au collision simulation). It seems to me that this is material dependent: would it occur in a perfect gas? (As is seen in the original Richtmyer-Meshkov instability). I should think not, not least as a perfect gas would not maintain a surface with a vacuum interface, unlike a solid or liquid.*

Authors – The same-material case is indeed surprising at first, however as presented in the paper, due to its geometric origin, the impact-induced instability appears even if the impacting bodies are made of the same material. Theoretically, this instability will also occur in the case of perfect gases, however, as the reviewer stated, it will be difficult to maintain the initial form of the interface.

2. *On page 6 I am not sure I understand the reasoning behind the choice of setting the post-impact Atwood number $At+$ to 1. Why is that value chose? If the materials are the same, then why is ρ_{02} difference from ρ_{01} ? Or are they somehow different due to the obliqueness of the shocks?*

Authors – We basically used Atwood number = 1 in order to eliminate it from the impulsive growth rate equation for the impact case. We added a short explanation on page 6.

3. *P1: "The vorticity is generated due to the corrugated initial interface causes the initial topography to grow over time" -- the sentence is unclear, I suggest inserting the word "that" before "causes..."*

Authors – we fixed the sentence (p. 1).

4. *P2: "This place the low-impedance material" -- should be "places" I think.*

Authors – we fixed the sentence (p. 2).

5. P6: The wording "sizes of shocked zone" is confusing since it seems to refer to a velocity difference (ΔU_s); "sizes" and "perturbation zone" makes me think the authors are talking about physical sizes (distances). Reading it over, it becomes even less clear to me: "relative mixing-zone size": What is the physical region that the authors are referring to? Relative to what? Impactor diameter?

Authors – Because we obtain constant growth velocity of the mixing zone and constant velocity growth of shocked zone, the ratio $v_{mix}/\Delta U_s$ is kept constant for the impact case at all times post impact. We rephrased this point in the text to make it clearer for the reader (p. 6).

Reviewers' Comments:

Reviewer #1:

Remarks to the Author:

The revision is fine. It addresses all the points raised by both reviewers and the result is a clearer manuscript. It is suitable for publication with no further corrections.

Reviewer #2:

Remarks to the Author:

Looking at the authors' responses to the reviews, and the revised manuscript, I believe that they have responded to the comments and revised the paper in a satisfactory manner. I thus am able to recommend publication.